# Quantitative Inversion Method of Surface Suspended Sand Concentration in Yangtze Estuary Based on Selected Hyperspectral Remote Sensing Bands

**Kuifeng Luan** [1,2]**, Hui Li** [1,]*****, Jie Wang** [1,2]**, Chunmei Gao** [3]**, Yujia Pan** [4]**, Weidong Zhu** [1,2]**, Hang Xu** [1]**, Zhenge Qiu** [1,2] **and Cheng Qiu** [4]

1  College of Marine Sciences, Shanghai Ocean University, Shanghai 201306, China
2  Estuarine and Oceanographic Mapping Engineering Research Center of Shanghai, Shanghai 200123, China
3  College of Marine Ecology and Environment, Shanghai Ocean University, Shanghai 201306, China
4  Shanghai Marine Monitoring and Forecasting Center, Shanghai 200062, China
*  Correspondence: adalyn0125@163.com

**Abstract:** The distribution of the surface suspended sand concentration (SSSC) in the Yangtze River estuary is extremely complex. Therefore, effective methods are needed to improve the efficiency and accuracy of SSSC inversion. Hyperspectral remote sensing technology provides an effective technical means of accurately monitoring and quantitatively inverting SSSC. In this study, a new framework for the accurate inversion of the SSSC in the Yangtze River estuary using hyperspectral remote sensing is proposed. First, we quantitatively simulated water bodies with different SSSCs using sediment samples from the Yangtze River estuary, and analyzed the spectral characteristics of water bodies with different SSSCs. On this basis, we compared six spectral transformation forms, and selected the first derivative (FD) transformation as the optimal spectral transformation form. Subsequently, we compared two feature band extraction methods: the successive projections algorithm (SPA) and the competitive adaptive reweighted sampling (CARS) method. Then, the partial least squares regression (PLSR) model and back propagation (BP) neural network model were constructed. The BP neural network model was determined as the best inversion model. The new FD-CARS-BP framework was applied to the airborne hyperspectral data of the Yangtze estuary, with $R^2$ of 0.9203, RPD of 4.5697, RMSE of 0.0339 kg/m$^3$, and RMSE% of 8.55%, which are markedly higher than those of other framework combination forms, further verifying the effectiveness of the FD-CARS-BP framework in the quantitative inversion process of SSSC in the Yangtze estuary.

**Keywords:** surface suspended sand concentration; first derivative; competitive adaptive reweighted sampling; neural network; feature band extraction; hyperspectral remote sensing

## 1. Introduction

Estuarine coastal areas are partially enclosed water bodies where the majority of land–sea interactions occur, and various processes are coupled and are associated with complex evolutionary mechanisms. Sediment movement plays a linking role [1] among these interactions, making the study of sediment movement patterns in estuarine coastal areas a common concern in related disciplines. Suspended sediment is an important water color element and water quality parameter that constitutes the spectral characteristics of estuarine and near-shore water bodies. Its content directly affects the water quality and optical properties of water bodies, such as turbidity and transparency [2]. The distribution of suspended sediment in the surface layer of estuaries is a specific reflection of estuarine sediment movement, and many scholars have studied the distribution dynamics of suspended sediment along estuarine coasts and their dispersion and transport patterns to different degrees [3–6]. The Yangtze River is the largest river in China, with $8.88 \times 10^{11}$ m$^3$ of water and $3.76 \times 10^8$ t of sediment transported into the East China Sea by the South

Channel, North Channel, North Port, and North Branch every year. The large amount of water and sand discharge has a significant impact on water temperature, sedimentation, and the ecological environment, not only in the Yangtze River estuary, but also in the adjacent shelf seas [7]. The Yangtze River estuary is one of the three major estuaries in China; it is a medium tidal estuary with branching estuaries and four mouths into the sea [8]. The complex topographic and hydrodynamic conditions cause a more complex distribution of suspended sand concentration. Therefore, the study of suspended sediment content and distribution in the Yangtze estuary has important scientific significance and application value.

In recent years, many scholars at home and abroad have carried out many studies based on remote sensing methods. Remote sensing inversion is characterized by the ability to achieve large-scale simultaneous observation and periodicity, and the inversion of suspended sediment concentration in the surface layer of estuarine coast by remote sensing means has become an important research tool. Remote sensing tools applied to the quantitative inversion of SSSC include multispectral and hyperspectral remote sensing. Studies based on multispectral remote sensing mainly use satellite images as data sources, including Landsat (Landsat MSS/TM/ETM+/OIL) [9–11], meteorological satellites (NOAA, FY) [12,13], and water color satellites (SeaWIFS, MERIS, GOCI, etc.) [14–17]. However, multispectral remote sensing has a smaller number of spectral bands and lower resolution. Hyperspectral remote sensing with its nanometer-scale spectral resolution can not only distinguish different water body types, but also better capture the spectral characteristics of water bodies for accurate inversion of SSSC [18]. Yang [19] used in situ hyperspectral data and corresponding water chemistry data from 7–8 March to 6–7 July, 20 September, and 7–8 December 2004, to establish regression algorithms for water quality parameters. Their results show that the peak water ionization radiance (R 700) at approximately 700 nm varies proportionally with chlorophyll-a (chl-a) concentration and shifts to the infrared when algal blooms occur. Wang [20] investigated the potential of satellite hyperspectral data, i.e., Hyperion images, for mapping the total suspended solids (TSS) concentration of coastal water in Liaodong Bay, China. After processing and atmospheric correction, the reflectance of the water extracted from the Hyperion images was used to express the spectral characteristics of different TSS concentrations. Gao [21] used in situ hyperspectral data and TSM concentration data for Changdang Lake, China, to establish a TSM concentration inversion model. The model was applied using 60 Sentinel-2 images acquired from 2016 to 2021 to determine the temporal and spatial distribution of TSM concentrations. Kwon [22] developed a robust machine learning (ML) model for SSC estimation based on hyperspectral images by considering the optical variability of suspended sediments in the water column.

Hyperspectral data possess the characteristics of large data volume and high covariance between bands, which leads to the problems of large computational volume, complex model structure, and poor stability of the inverse model of SSSC constructed using full bands or bands with high correlation [23,24]. Therefore, the effective selection of characteristic bands is a key issue in constructing a robust hyperspectral inverse model. In the study of SSC inversion, the commonly used method is correlation analysis, that is, the correlation coefficient between the spectral reflectance and the measured SSSC is calculated, and the spectral band corresponding to the high correlation coefficient is defined as the sensitive band. Then, the inversion model is built. Based on Landsat 5 thematic map (TM) images and a set of field datasets, J Kong [25] developed a reliable and sensitive inversion model for SSC levels in the Caofeidian area of the new seaport in northeastern China, and selected a sensitive waveband for the model by calculating correlation coefficients. Womber Zelalem R. [26] et al. conducted a correlation analysis using MODIS-Terra and measured SSC data, and found that MODIS-Terra reflectance correlated best with measured SSSC in the near-infrared band. However, the correlation analysis did not consider interband correlation, and there were still problems of covariance and redundancy between the characteristic bands. Feature band selection methods, such as the successive projections algorithm (SPA) and the competitive adaptive reweighted sampling (CARS) method, can

effectively eliminate the influence of covariance among many wavelength variables and reduce the complexity of the model. They have been increasingly applied owing to their simplicity and speed, and have achieved good results in feature band extraction. Goudarzi et al. [27] used SPA to select the feature bands and compared it with a genetic algorithm (GA) to construct a PLSR model to predict the octanol/water partition ratio coefficients of 10 selected halogenated benzoic acids. Fei et al. [28] proposed a continuous projection algorithm-least squares support vector machine (SPA-LS-SVM) framework for the prediction of acetic, tartaric, and lactic acids in plum vinegar, and found that SPA resembled the correlation coefficient method. Wei et al. [29] proposed a deep neural network with CARS (DNN-CARS) to estimate the content and spatial distribution of abrupt TAs. T Wu et al. [30] used the CARS and SPA for band selection and built a multiple linear regression based on the characteristic bands model to predict soil water content, and found that the accuracy of prediction was high. However, relatively few studies have applied SPA and CARS to the quantitative inversion of SSSC.

Constructing a robust hyperspectral inversion model based on an effective feature band is another key issue in the quantitative inversion of SSC. Methods applied in remote sensing inversion studies of SSSC are mainly divided into empirical [31–34], semi-empirical [16,17,35,36], and analytical [37,38]. Among them, the empirical method is simple and reliable, but relies on a large amount of actual measurement data and has poor generality; the semi-empirical method has the advantage of strong physical correlation of the analytical method and the characteristics of operability of the empirical method. It is the most widely used remote sensing quantitative inversion method; however, its practicality is reduced due to its dependence on actual water surface measurement data and its synchronization with remote sensing data [35,36]. The analytical method is the most advanced, and can achieve accurate inversion of SSSC, although it requires methodological iterations and experience [37,38]. In recent years, with the increasing spectral, temporal, and spatial resolution of remote sensing data, as well as the increasing research on water color remote sensing, scholars have started to search for more accurate and generalizable models. Partial least squares (PLS) is a multivariate statistical regression analysis that realizes dimensionality reduction of remote sensing data by establishing a regression model between variable datasets [39]. To assess the feasibility of using reflectance spectroscopy to map the abundance of soil Pb and other heavy metals, Pandit [40] investigated the relationship between surface soil metal concentrations and hyperspectral reflectance measurements using partial least squares regression (PLSR) modeling. Axelsson [37] compared the performance of a support vector regression (SVR)-based model with a partial least squares regression (PLSR)-based model with regards to the possibility of recovering nitrogen, phosphorus, potassium, calcium, magnesium, and sodium concentrations from mangrove forests in the Belau Delta, Indonesia; Lu [41] and Song [42] used a genetic algorithm (GA) to perform PLS inversion of chl-a concentration in Shikoumen Reservoir after the preferential selection of bands, band ratios, and first-order differentiation that correlate well with chl-a concentration, and achieved high modeling and prediction accuracy. In addition, many studies have shown that neural network algorithms have strong vitality in fitting nonlinear relationships, and that their self-organization, self-learning characteristics, and strong fault tolerance have unique advantages in remote sensing simulation and prediction of water quality parameters [43,44]. Among the many neural network algorithms, the BP neural network model is the most common. Samli [45] developed a back-propagation neural network (BP-ANN) model for estimating the chl-a concentration from the obtained input values using an ANN structure consisting of three input neurons and one output neuron. Based on the Landsat 8 remote sensing image data of Wuliangye Lake and the measured chl-a concentration sampling points of Wuliangye Lake, Fu [46] constructed 26 BP neural network models to retrieve the chl-a concentration of Wuliangye Lake, using the first to fifth band spectral reflectance combination of Landsat 8 remote sensing image data as the input and the measured chl-a concentration as the output.

Most of the current studies focus on a single problem of feature band extraction or inversion model building, and there are few studies on improving the inversion accuracy of the whole technical process of "spectral preprocessing–feature band extraction–inversion model building". In this study, the SSSC was quantitatively simulated using sediment samples from the Yangtze estuary region. First, the spectral characteristics of water bodies with different SSCs were analyzed. Second, this study combines six spectral transformation forms; extracts feature bands using SPA and CARS methods; builds a PLSR model and BP neural network model based on feature bands; calculates four evaluation indices, $R^2$, RPD, RMSE, and RMSE%, to determine the optimal spectral transformation form, feature band extraction, and inversion model building methods based on hyperspectral data in the inversion process of SSSC in the Yangtze estuary; and constructs the best framework combination form. The best framework combination was constructed by determining the optimal spectral transformation, feature band extraction method, and inversion model building method. Finally, the constructed framework was applied to the 2016 airborne hyperspectral simultaneous monitoring experiment in the Yangtze River estuary to further validate its effectiveness in the inversion process of the SSSC in the Yangtze River estuary, and to provide methodological support for the hyperspectral remote sensing inversion of the SSSC.

## 2. Materials

### 2.1. Quantitative Experimental Data and Preprocessing

The experiment for the quantitative determination of the SSSC was conducted in a white plastic tank with a length and width of 106 cm and a height of 115 cm. Before the experiment, the interior of the tank was painted black to eliminate the effect of solar reflection on the bottom and inner walls of the tank. The water was stirred uniformly using an electric stirrer to ensure that the sediment was suspended evenly in the water column.

The surface sediment samples needed for the quantitative experiments were collected using a sediment sampler near the south sink mouth of the Yangtze River estuary (30°55′44″ N, 121°59′36″ E). The sample particle size was determined using a Master-sizer 2000 laser particle size analyzer, and the median particle size (D50) was 28.63 μm, which matched the particle size of the suspended sediment in the surface layer of the Yangtze estuary [47]. Sediment samples were weighed and stored in plastic bags for further use in subsequent experiments. Spectral reflectance was measured using an ASD Field Spec portable hyperspectrometer with the parameters listed in Table 1.

**Table 1.** Parameters of the ASD hyperspectrometer.

| Parameters | Value |
| --- | --- |
| Wavelength Range | 325–1075 nm |
| Sampling Interval | 1 nm |
| Spectral Resolution | 3 nm |
| Integration time | ≥8.5 ms |
| Field of view | 25° |

The experiment was conducted on 19 January 2021, from 10:00 to 14:30 in an open area on the campus of Shanghai Ocean University under clear and cloudless weather and good lighting conditions. A bucket was filled with 1000 L of tap water and sediment samples were added sequentially in ascending order. After each addition, the samples were stirred with an electric stirrer to ensure a uniform suspension of sediment samples in the water column. After stirring for approximately 5 min, an AQU Alogger 310TY turbidimeter with a range of 10,000 FTU was used for online observation of turbidity. When the values were stable, the spectral characteristics of the SSSC were measured using the above-water spectrum measurement method [48]. The reflectance of the standard plate, sky, and sand-bearing water bodies was measured using an ASD Field Spec portable hyperspectrometer, calibrated by a standard plate with 20% reflectance, and 30 consecutive

spectral data points were collected for each group. The hyperspectrometer outputs were observed from the water surface at a zenith angle of 40° and in real time at an azimuth of 135°, effectively avoiding the reflective effects of solar irradiance [49]. Water samples were collected at a depth of 0.5 m using a standard sampling bottle with a capacity of 1 L, while obtaining spectral information of the water. The water samples were also brought back to the laboratory, and the SSSC information was determined by the weighing method. A total of 41 sets of spectral reflectance information for sand-bearing water bodies were obtained. The experimental site is shown in Figure 1.

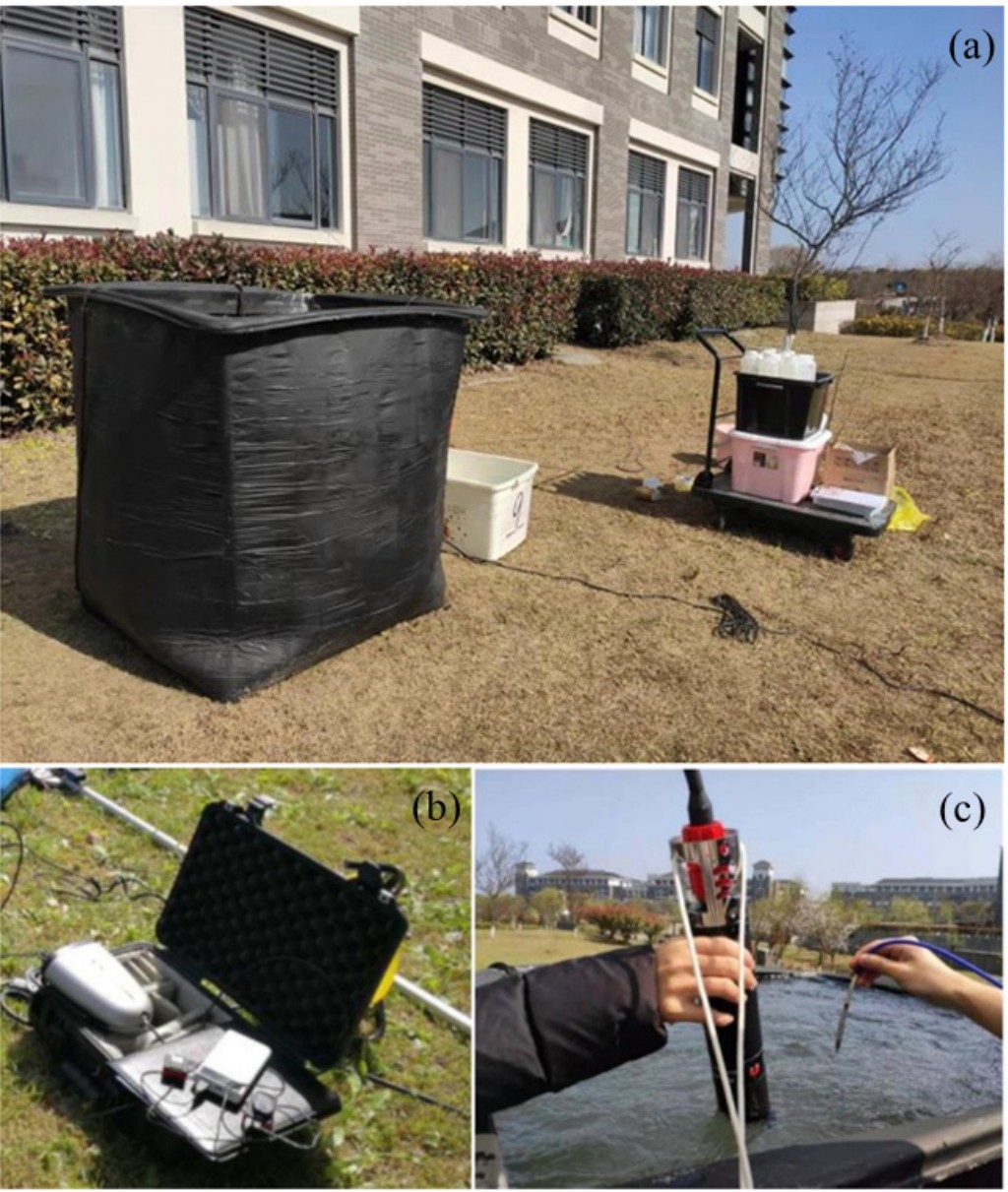

**Figure 1.** Quantitative experimental scenes and main instrumentations. (**a**) Quantitative experimental scenes. (**b**) ASD Field Spec handheld hyperspectrometer. (**c**) AQU Alogger 310TY turbidimeter with a range of 10,000 FTU.

The spectral reflectance of water bodies with different SSSCs can be calculated from the reflectance data acquired by the ASD Field Spec hyperspectrometer; its spectral reflectance $R_{rs}$ is determined using Equation (1).

$$R_{rs} = \frac{(S_{sw} - r \times S_{sky}) \times \rho_P}{\pi \times S_P} \tag{1}$$

where $S_p$, $S_{sw}$, and $S_{sky}$ are the average reflectance measurements of the standard plate, sand-bearing water body, and sky, respectively; $\rho_p$ is the reflectance of the standard plate; and *r* is the reflectance of the water–air interface. In this experiment, the wind speed was approximately 5 m/s and the value of r was determined to be 0.025 [49].

The actual measured concentration of the suspended sand was determined using laboratory methods. The specific steps were as follows: The water samples were first shaken to ensure they were well mixed; the water samples were then filtered using 0.45 μm glass fiber filter membranes. The membrane equilibration operation was performed for 6 h before filtration (to prevent the effect of difference in humidity on the weight of the membrane), and the membrane itself was weighed using an electronic balance of 10,000 parts. According to the filtration method specifications, the water sample bottles were rinsed using ultrapure water, and the filter membranes were dried and equilibrated for 6 h after filtration before weighing [50]. Their surface suspended sand concentration SSSC was calculated as follows:

$$SSSC = \frac{M - M_0}{V} \qquad (2)$$

where *M* is the mass of the filter membrane after weighing, $M_0$ is the mass of the membrane before weighing, and *V* is the water sample volume.

## 2.2. Airborne Hyperspectral Experiment Data and Preprocessing

The airborne hyperspectral experiment was carried out upstream of the northern port of the Yangtze estuary (Figure 2) during the dry water period on 26 March 2016. The Yangtze River estuary diverges from below Xu Liujing to the northern and southern sides of Chongming Island, which are the South and North Branches, respectively. The South Branch diverges into the southern and northern harbors at Changxing Island and Heng Sha Island, and the South Harbor diverges into the South and North Troughs at Jiu Duan Sha [51], showing a "three-stage branching and four-port entry" pattern. The North Port is one of the four sea entry channels, with a total length of approximately 32 km.

Airborne hyperspectral and ground truth sampling data were acquired during the experiment. The airborne hyperspectral data were obtained using the hyperspectral camera onboard the manned aircraft. The imaging device in the experiment is an airborne hyperspectral sensor developed by the Shanghai Institute of Technology Physics, Chinese Academy of Sciences, the main parameters of which are shown in Table 2, and the specific parameters are listed in Table 2. The location information of the sampling points for the ground truth sampling data is shown in Figure 2, where sampling points 1–5 were taken from the shore using a rope, sampling points 6–14 were taken on a boat, and the sampling time was concentrated around 10:00–13:00, with good weather conditions and a calm water surface.

**Table 2.** Parameters of the airborne hyperspectrometer.

| Parameters | Value |
| --- | --- |
| Wavelength Range | 300–1000 nm |
| Number of Channels | 270 |
| Spectral Resolution | 2.6 nm |
| Flight altitude | 1000 m |
| Spatial resolution | 1.2 m |

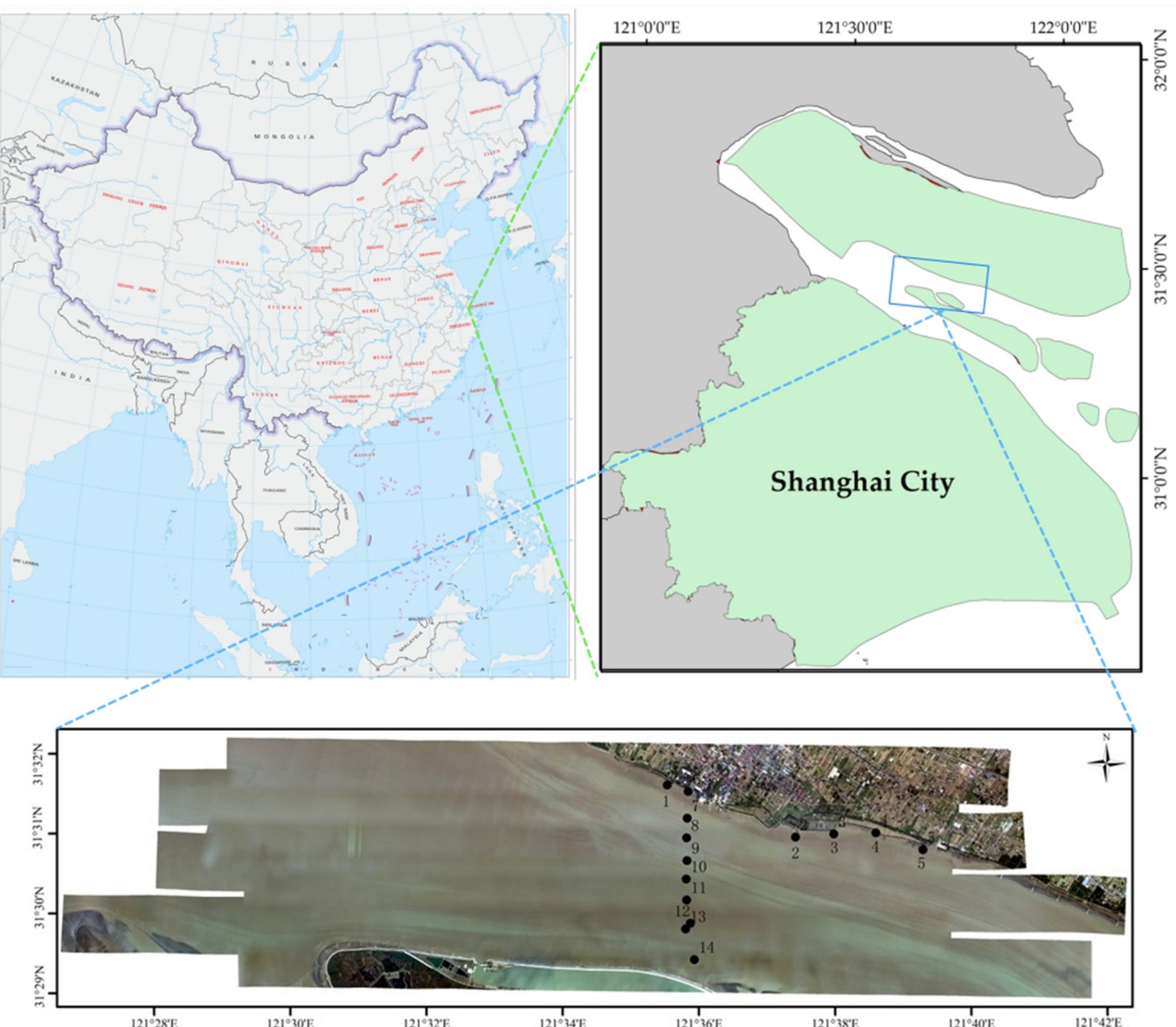

**Figure 2.** Airborne hyperspectral experiment area. The green square on this figure is the location of Shanghai. The green line leads to a magnified view of the area. The blue square on this figure is the area covered by the airborne hyperspectral image. The blue line leads to the hyperspectral image map and the location map of the sampling points.

The hyperspectral images were preprocessed using ENVI software, including geometric correction, image stitching, band removal, and image fusion, to obtain processed image data. During image preprocessing, 100 bands with low signal-to-noise ratios were removed to ensure data quality, and 170 bands were retained, with the remaining bands ranging from 420–900 nm. The processing method of the collected surface water samples was consistent with that of the water samples measured from the quantitative experiments. The final results of the SSSC data are listed in Table 3.

**Table 3.** Measured SSSC.

| ID | SSSC (mg/L) | ID | SSSC (mg/L) |
|----|----|----|----|
| 1 | 554.5 | 10 | 197.6 |
| 2 | 451.5 | 11 | 259.6 |
| 3 | 809.8 | 12 | 420.3 |
| 4 | 843.6 | 13 | 476.6 |
| 5 | 968.8 | 14 | 439.5 |
| 6 | 729.5 | Max | 968.8 |
| 7 | 357.8 | Min | 197.6 |
| 8 | 345.3 | Mean | 511.8 |
| 9 | 310.9 | SD | 288.3 |

## 3. Methods

### 3.1. Spectral Transformation Methods

The spectral transformation of the raw spectrum is based on five spectral transformation methods, and the equations of the spectral transformation methods are shown in Table 4. The raw spectra and five spectral transformation values were used for the construction of the quantitative inversion model of SSSC, and these transformation forms were implemented in the MATLAB platform.

**Table 4.** Spectral transformation methods.

| Name | Method or Formula | Abbreviation |
|----|----|----|
| First Derivative | Savitzky–Golay method | FD |
| Second Derivative | Savitzky–Golay method | SD |
| Square Root | $x_{sqrt} = \sqrt{x_i}$ | SQR |
| Mean Centering | $x_{mcenter} = x - \overline{X}$ | MC |
| Reciprocal of Logarithmic | $x_1/lg = 1/\log_{10}(x_i)$ | RLG |

### 3.2. Feature Band Extraction Methods

#### 3.2.1. Successive Projections Algorithm

The successive projection algorithm (SPA) is a forward selection method; that is, it starts with one band and selects a new band in each iteration until the specified number of feature bands, N, is reached, thus extracting the subset of feature variables with the least redundancy and covariance. The basic principle is to construct new variables by projecting and mapping spectral information and evaluating the predictive effect of the new variables based on a multiple linear regression model [52]. The SPA extracts a few columns of data from the original spectral matrix to aggregate the spectral variable information of most samples, thus avoiding the problem of data redundancy to the greatest extent. Moreover, the number of variables in the model building process can be reduced considerably, thereby improving the accuracy and efficiency of the model. The core formula is as follows [53]:

$$Px_j = x_j - (x_j^T x_{k(n-1)}) x_{k(n-1)} \left( x_{k(n-1)}^T x_{k(n-1)} \right)^{-1} \tag{3}$$

where $P$ is the projection operator, $j \in S$, and $S$ is the set of wavelengths not yet selected. $K$ represents the selected wavelength.

#### 3.2.2. Competitive Adaptive Reweighted Sampling Algorithm

The competitive adaptive reweighted sampling (CARS) method is based on the principle of "survival of the fittest" for feature band selection. Each band variable is treated as a single individual, and the individual with strong adaptive ability is retained for individual selection [54,55]. The specific steps of the algorithm are as follows:

(1)　Random selection of *n* samples using a Monte Carlo algorithm and the development of a partial least squares regression (PLSR) model.

(2)　Selecting the variables by exponentially decreasing function (EDP) and adaptive weighted sampling algorithm (ARS), retaining those with high regression coefficients and removing those with low regression coefficients.

(3)　Create a PLSR model with the retained variables as a new subset of variables and calculate the root mean square error of cross-validation (RMSECV).

(4)　Repeat steps (1)–(3), and select N subsets of variables after N Monte Carlo sampling to obtain N RMSECVs, and select the subset of variables with the smallest RMSECV as the optimal band combination.

### 3.3. BP Neural Network

The learning process of the BP neural network included forward and backward transmissions. In the forward transmission process, the signal enters the hidden layer through the input layer, is processed layer-by-layer by the hidden layer node, and is transmitted to the output layer. Each layer of neural nodes in the hidden layer only affects the state of the next layer of the neural nodes. When the processed signal reaches the output layer, the output layer evaluates the processing results of the hidden layer. If the output layer does not obtain the expected output, the network returns to transmit an error signal along the original connection channel. Each node in the hidden layer adjusts the weight of each layer of the neural nodes through the error information feedback, thereby minimizing the error signal [56].

The BP neural network model used in this study for predicting the concentration of suspended sand is shown in Figure 3.

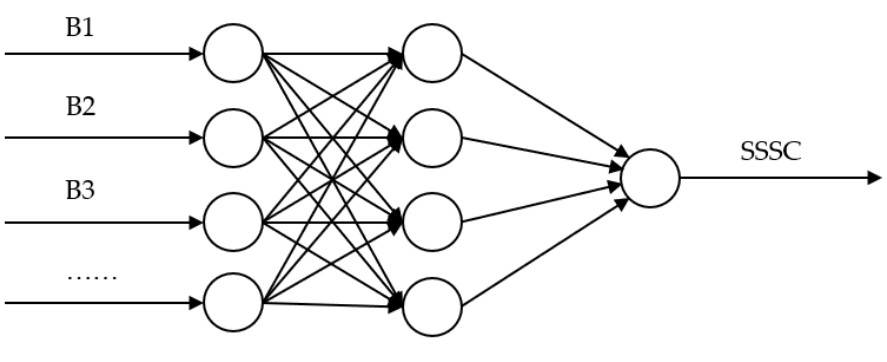

**Figure 3.** Prediction model of sand content based on BP algorithm.

### 3.4. Model Evaluation Indices

The four evaluation indicators of coefficient of determination ($R^2$), ratio of performance deviation (*RPD*), root mean square error (*RMSE*), and root mean square error percentage (*RMSE*%) were used to evaluate the performance of the inverse model of suspended sand concentration, and are calculated as follows:

$$R^2 = 1 - \frac{\sum_{i=1}^{N} (\gamma_i - \hat{\gamma}_i)^2}{\sum_{i=1}^{N} (\gamma_i - \overline{\gamma}_i)^2} \tag{4}$$

$$RPD = \frac{SDs}{RMSE}$$

$$RMSE = \sqrt{\sum_{i=1}^{N} \frac{(\gamma_i - \hat{\gamma}_i)^2}{N}}$$

$$RMSE\% = \frac{RMSE}{\overline{\gamma}_P}$$

where $N$ is the number of samples, $\gamma_i$ is the measured value of $SSSC$, $\hat{\gamma}_i$ is the predicted value of $SSSC$, $\overline{\gamma}_i$ is the average measured value of $SSSC$, and $\overline{\gamma}_P$ is the average predicted value of $SSSC$.

$R^2$ indicates the strength of the correlation between measured and predicted values; $RPD$ indicates the predictive ability of the model, and $RPD$ less than 1.5 indicates very poor predictive ability of the model, between 1.5 and 2.0 indicates poor predictive ability of the model, and greater than 2.0 indicates good predictive ability of the model [57]. $RMSE$ indicates the standard deviation of the prediction error, and $RMSE\%$ is the percentage of the standard deviation. Smaller $RMSE$ and $RMSE\%$ values indicate a higher prediction accuracy of the model.

## 4. Results

### 4.1. Quantitative Experimental Spectral Characteristic Curve

A total of 41 sets of measured $SSSC$ data were obtained, with a minimum value of 3.62 mg/L, a maximum value of 682.06 mg/L, and an average value of 203.07 mg/L. During the experiments, the spectral data of 41 sets of water bodies with different SSSCs were obtained simultaneously, and spectral characteristic curves were drawn, as shown in Figure 4. The spectral ranges of 325–400 nm and 900–1075 nm in the measured spectral information are susceptible to external interference, and have a low signal-to-noise ratio due to the high absorption properties of the water column [23]. Therefore, the spectral range in this study was 400–900 nm. It was observed that when the concentration of suspended sand is low, the spectral curve has one peak at 560–580 nm; an increase in the concentration of suspended sand leads to the spectral curve of the water body having two peaks: the first reflection peak is located at 570–710 nm, the reflection peak is flatter and corresponds to a wider wavelength range; when the wavelength is greater than 710 nm, the spectral curve begins to decline and a reflection valley is formed at 750 nm, mainly due to the absorption of water molecules, followed by a second reflection peak at 780–820 nm, which has a narrower width.

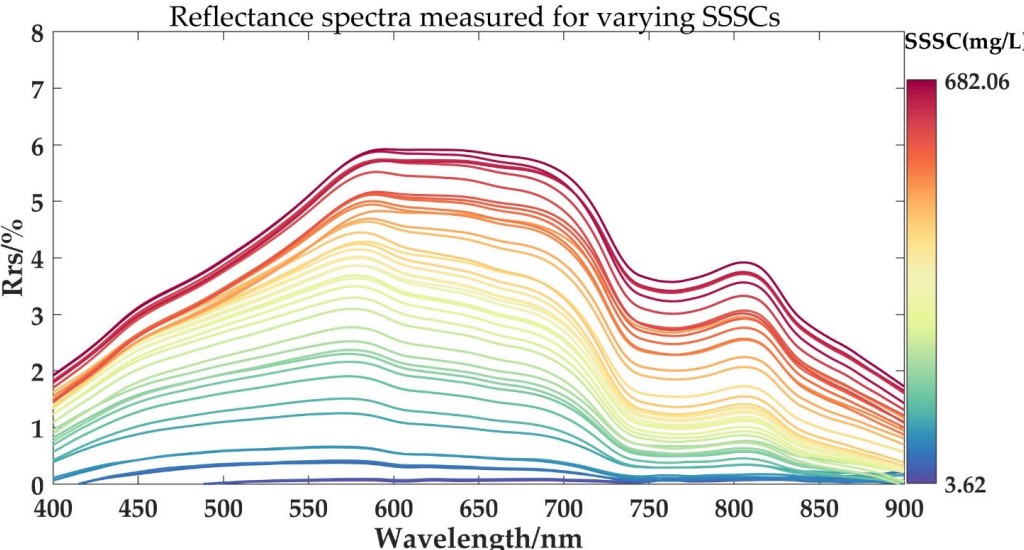

**Figure 4.** Original spectral characteristic curves of different SSSCs.

Five spectral transformations were performed on the raw spectra, and six different spectral expressions were obtained, as shown in Figure 4. It was observed that, for the raw spectra, the spectral reflectance of each band shows different degrees of an increasing trend with the increase in SSSC, and the spectral characteristic curves of sand-containing water bodies have similar morphologies, but there are also differences. FD mainly shows

the changing trend of the raw spectral curve in the wavelength range of 400–563 nm and 763–805 nm, indicating that FD is positive. When it shows the changing trend of the raw spectral curve in the range of 595–762 nm and 807–900 nm, FD is negative, indicating a decreasing trend in the original spectral curve, and in the range of 607–679 nm, FD is close to zero, indicating a flatter original spectral curve (Figure 5b). The spectral transformation value of SD is further reduced compared with that of FD, and the reflectance transformation value revolves around the upper and lower $Y = 0$ (Figure 5c). The spectral transformation form of the MC did not change the morphology of the spectral curve, but only the value of the $y$ axis reflectance (Figure 5f). The morphology of the spectral curve of SQR was similar to that of the spectral transformation form of RS, but the magnitude of the change was slightly different, especially in the low concentration range (Figure 5d). The morphology of the spectral curve of RLG was exactly opposite, but in the low concentration range, the spectral curve of RLG showed a clear peak around 500 nm (Figure 5e).

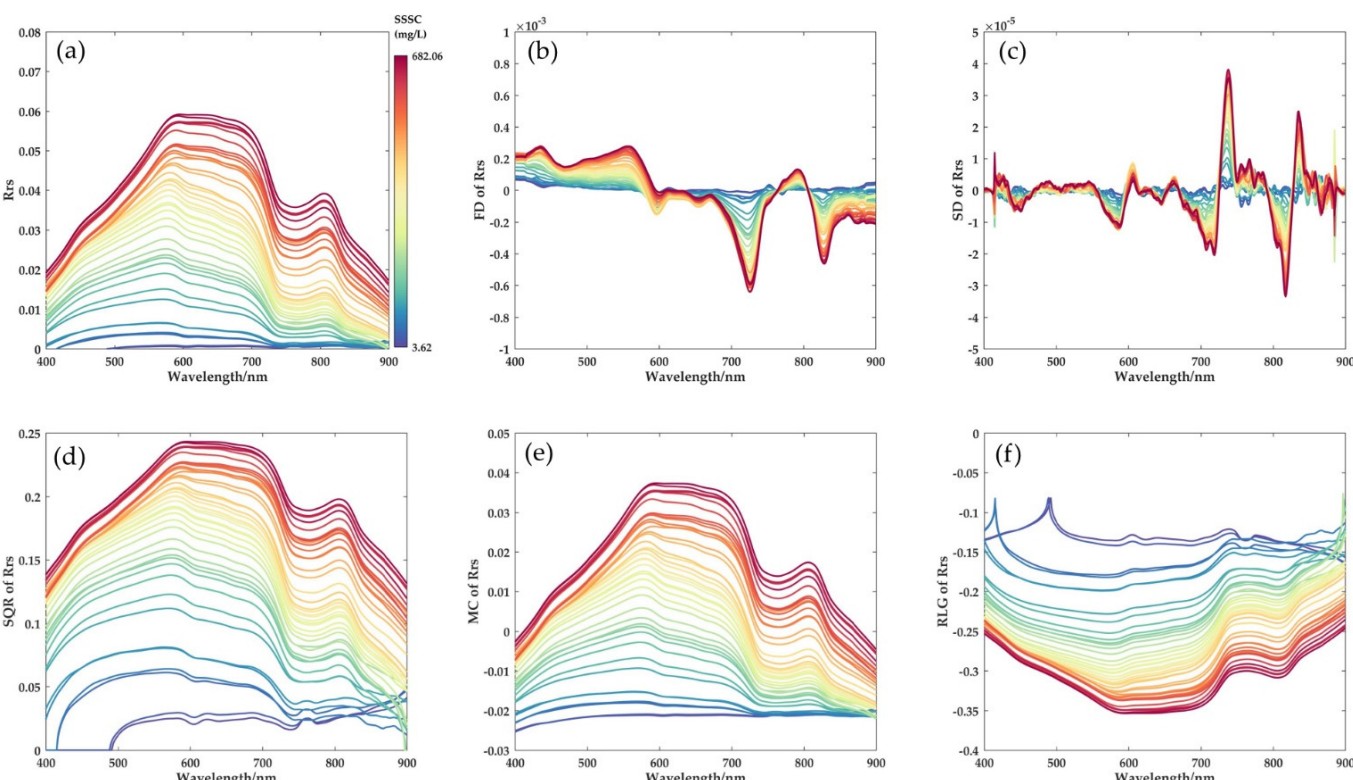

**Figure 5.** Spectral characteristic curves of 6 spectral transformation forms. (**a**) The raw reflectance spectra (RS), (**b**) the first derivative of the reflectance spectra (FD), (**c**) the second derivative of the reflectance spectra (SD), (**d**) the square root of the reflectance spectra (SQR), (**e**) the reciprocal of logarithm of reflectance spectra (RLG), (**f**) the mean center of the reflectance spectra (MC).

*4.2. Feature Band Extraction Results*

For the above six different spectral transformation values, 41 sets of sample data for each spectral transform value were divided into 30 sets of training samples and 11 sets of validation samples according to the concentration gradient method (Table 5). Thirty sets of training samples were used for SPA and CARS feature band extraction, and the extracted feature bands were used for modeling. Figure 6 shows the feature bands extracted by six spectral transform forms based on SPA. Most of the feature bands extracted by the SPA method are regions with significant changes, which indicates that the SPA method can effectively extract the feature bands [35]. Figure 7 shows the details of the feature bands extracted based on CARS, with FD as an example. Figure 7a shows the change in the number of selected spectral variables in 50 iterations. With the increase in the number of iterations, the number of spectra selected by CARS first decreases sharply and then levels

off. Figure 7b shows the change in RMSECV in 50 iterations. It was observed that with the increase in the number of iterations, RMSECV first shows a decreasing trend, then an increasing trend, and finally levels off. RMSECV reaches the minimum value in the 35th iteration. Figure 7c shows the path of regression coefficients for all spectral variables, i.e., the trend of regression coefficient values in 50 iterations. From Figure 7, it was observed that the spectral variables selected in the 34th iteration were the best subset of variables in the process of extracting feature bands based on CARS.

**Table 5.** Datasets for training and validation.

| Datasets | No. | Minimum (mg/L) | Maximum (mg/L) | Average (mg/L) |
|---|---|---|---|---|
| training | 30 | 3.64 | 654.39 | 200.39 |
| validation | 11 | 3.62 | 682.06 | 210.40 |

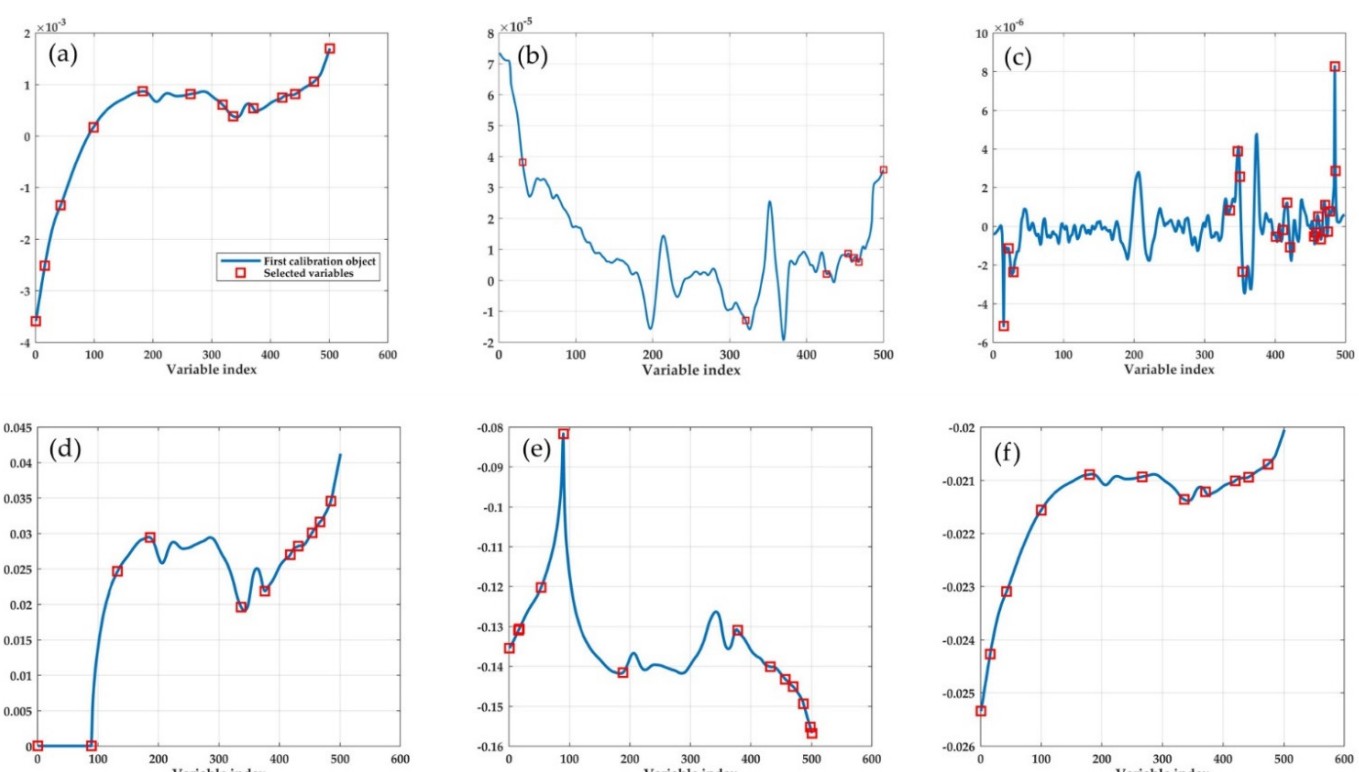

**Figure 6.** SPA−based feature bands extracted by different spectral transformation forms. (**a**) The raw reflectance spectra (RS), (**b**) the first derivative of the reflectance spectra (FD), (**c**) the second derivative of the reflectance spectra (SD), (**d**) the square root of the reflectance spectra (SQR), (**e**) the reciprocal of logarithm of reflectance spectra (RLG), (**f**) the mean center of the reflectance spectra (MC).

The six spectral transformation values were extracted using SPA and CARS for feature band extraction. Figure 8 shows the concentration diagram of feature bands selected based on the SPA and CARS feature band extraction methods, where the horizontal axis shows the different wavelength ranges, the vertical axis shows the six spectral transform forms, and the different colors represent the frequencies of spectral band selection. The results show that the feature bands selected by SPA and CARS are not exactly the same. First, the number of selected feature bands was different, and the number of feature bands selected by CARS was higher than that selected by SPA. Second, the distribution ranges of the selected feature bands are different, and the feature bands selected by SPA are concentrated at 400–450 nm and 850–900 nm, whereas the feature bands selected by CARS are different. CARS selects a wider range of feature bands than SPA, concentrating at 400–500 nm, 700–750 nm, and 800–900 nm. Overall, the feature bands selected by the two methods are distributed in the

visible band at approximately 400 nm and the near-infrared band at 700–900 nm, which is consistent with the findings of previous studies [58].

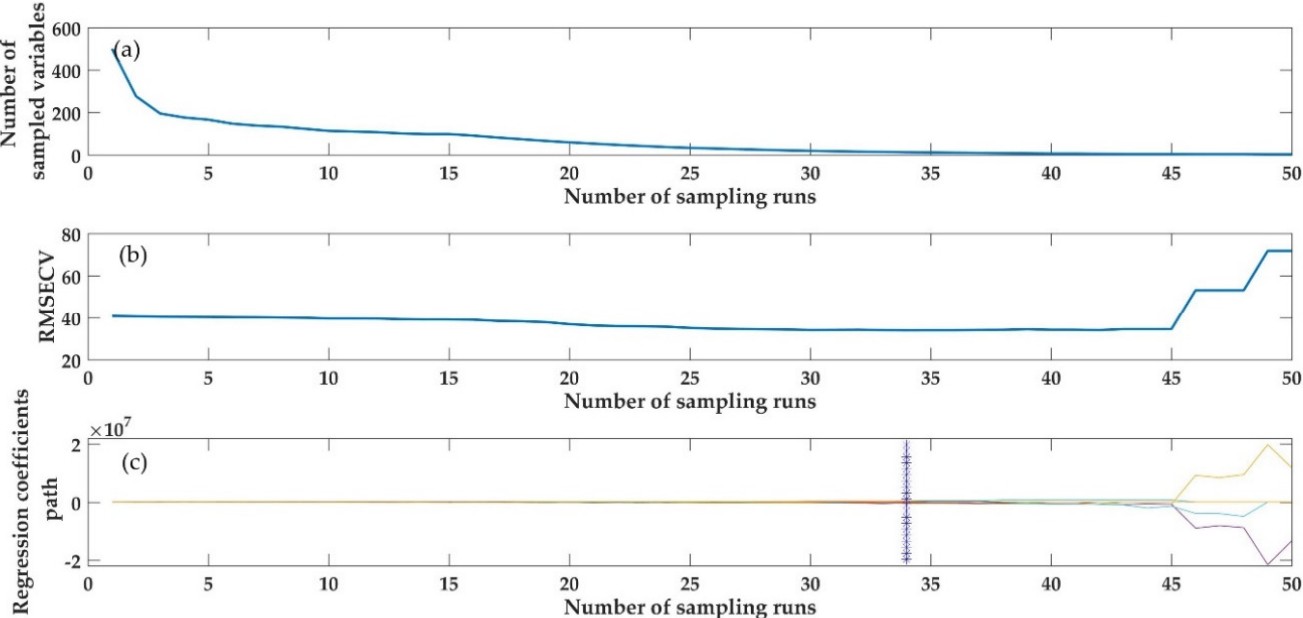

**Figure 7.** CARS feature band extraction process of FD. (**a**) The number of sampled variables, (**b**) 5-fold root mean squared error of cross−validation (RMSECV) values, and (**c**) regression coefficient path of each spectral variable during the 50 iterations.

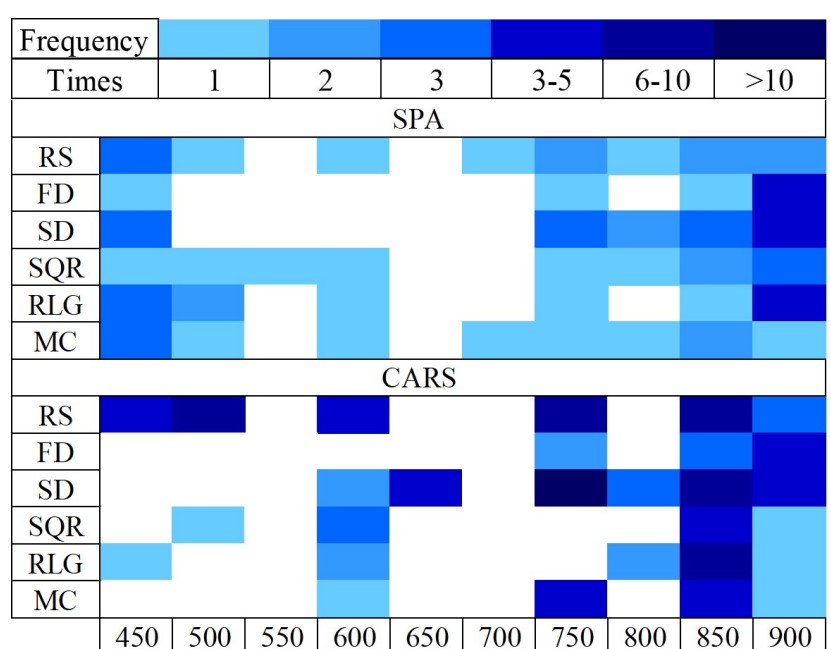

**Figure 8.** Concentration of feature bands extracted by SPA and CARS.

### 4.3. Construction of Inverse Model Results Based on Feature Bands

The PLSR model and BP neural network model were constructed by combining the six spectral transformation forms based on the feature bands extracted by SPA and CARS with the measured SSSC, and the model accuracy was evaluated using four indices, $R^2$, RPD, RMSE, and RMSE%, as shown in Figures 9 and 10. Figure 9 shows the results of inverse modeling based on SPA-extracted feature bands, and it was observed that the accuracies of the PLSR model and BP neural network model constructed based on different

spectral transformation forms are different. Comparing the $R^2$ values, it was observed that the highest $R^2$ is 0.9927 for the FD-based SPA-BP model, and the lowest $R^2$ is 0.9119 for the SD-based SPA-PLSR model; comparing the RPD, it was observed that the highest accuracy is achieved for the FD-based SPA-BP model, with an RPD of 10.5649, and the lowest accuracy is achieved for the SD-based SPA-PLSR model, with an RPD of 3.0364; similarly, comparing RMSE and RMSE% shows that the SPA-BP model constructed based on FD has the highest accuracy, with 20.9943 mg/L RMSE and 9.92% RMSE%, and the SPA-PLSR model constructed based on SD has the lowest accuracy, with 69.8012 mg/L RMSE and 29.47% RMSE%.

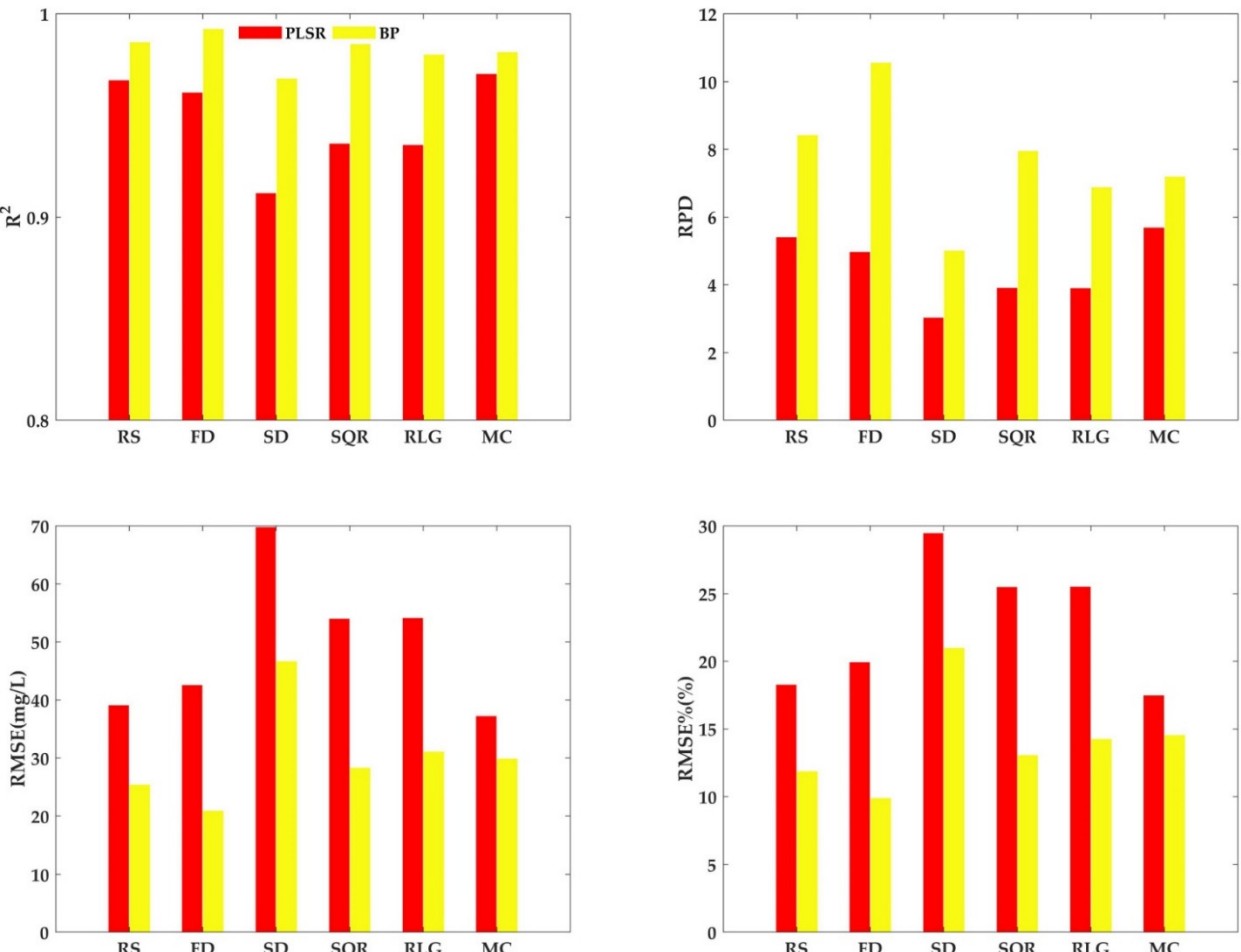

**Figure 9.** Results of inversion modeling based on SPA-extracted feature bands.

Figure 10 shows the results of inverse modeling based on CARS-extracted feature bands, where the accuracy of the PLSR model and BP neural network model constructed based on different spectral transformation forms is different. Comparing the $R^2$ values, it was observed that the highest $R^2$ is 0.9947 for the FD-based CARS-BP model, and the lowest $R^2$ is 0.9428 for the SQR-based CARS-PLSR model; comparing the RPD, it was observed that the highest accuracy is achieved by the FD-based CARS-BP model, with an RPD of 12.5453, and the lowest accuracy is achieved by the SQR-based CARS-PLSR model, with an RPD of 4.9796. The same comparison of RMSE and RMSE% shows that the CARS-BP model constructed based on FD has the highest accuracy, with an RMSE of 16.5801 mg/L and RMSE% of 8.08%, whereas the CARS-PLSR model constructed based on SQR had the lowest accuracy, with an RMSE of 52.1268 mg/L and RMSE% of 24.12%. Overall, FD is the best form of spectral transformation for both SPA and CARS to extract feature bands.

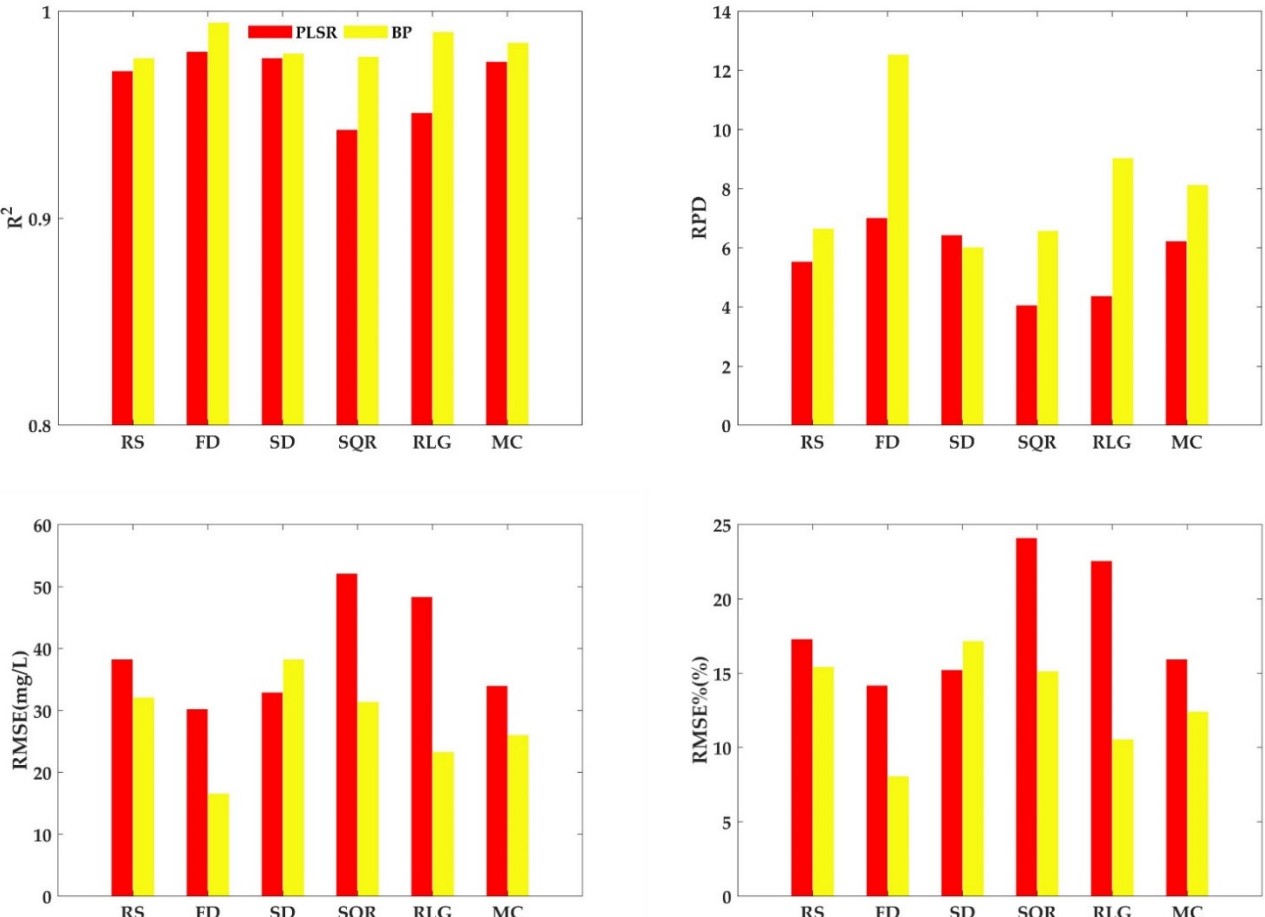

**Figure 10.** Results of inversion modeling based on CARS-extracted feature bands.

The accuracies of the SPA-PLSR/BP and CARS-PLSR/BP models with FD spectral transformation values are listed separately, as shown in Figure 11, and it was observed that for both SPA and CARS feature band extraction methods, regardless of the established PLSR model or BP neural network model, the accuracy of CARS is higher than that of SPA, specifically in terms of higher $R^2$ and RPD, and lower RMSE and RMSE%. For the same feature band extraction method, the accuracy of the BP neural network model is markedly higher than that of the PLSR model. Therefore, the best framework combination is FD-CARS-BP for the hyperspectral-based inversion of the SSSC in the Yangtze estuary.

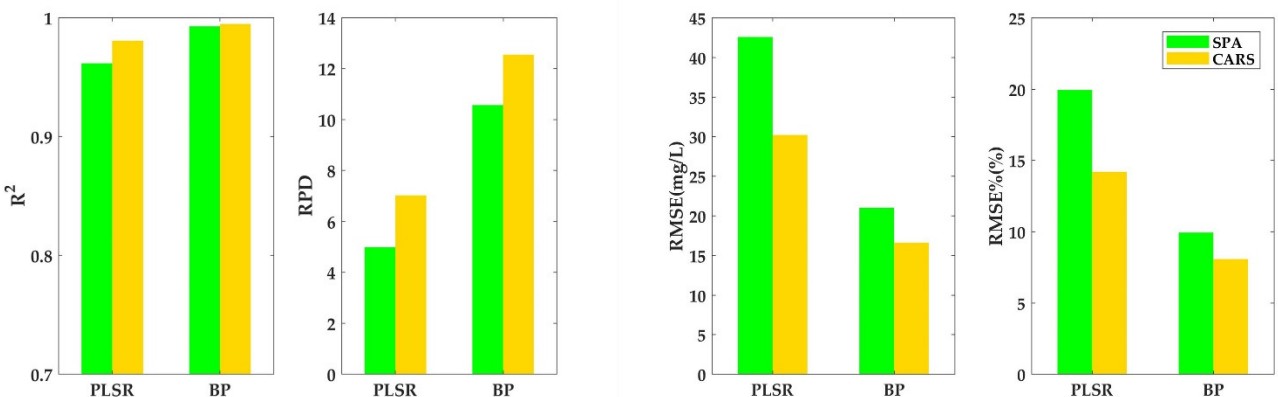

**Figure 11.** Accuracy of SPA-PLSR/BP and CARS-PLSR/BP models for FD.

### 4.4. Validation of the Model Framework

To verify the effectiveness of the best framework combination, FD-CARS-BP, obtained based on quantitative experiments in the inversion of SSSC in the Yangtze estuary, the framework was applied to the airborne hyperspectral data acquired on 26 March 2016. First, the first-order differential spectral transformation was performed on the airborne hyperspectral data to obtain the FD spectral transformation value, and then the feature bands were extracted based on CARS. The BP neural network model was built based on the feature bands, and $R^2$, RPD, RMSE, and RMSE% were calculated to evaluate the model accuracy. The FD-SPA-PLSR, FD-SPA-BP, and FD-CARS-PLSR models were also established to compare their inversion accuracies, as shown in Figure 12. Compared with the other three models, the $R^2$ and RPD of the FD-CARS-BP model were the largest, 0.9203 and 4.5697, respectively, and the RMSE and RMSE% of the FD-CARS-BP model were the smallest (0.0339 kg/m$^3$ and 8.55%, respectively), indicating that FD-CARS-BP can be effectively used for the extraction and inversion of the characteristic waveband of SSSC in Yangtze River estuary modeling.

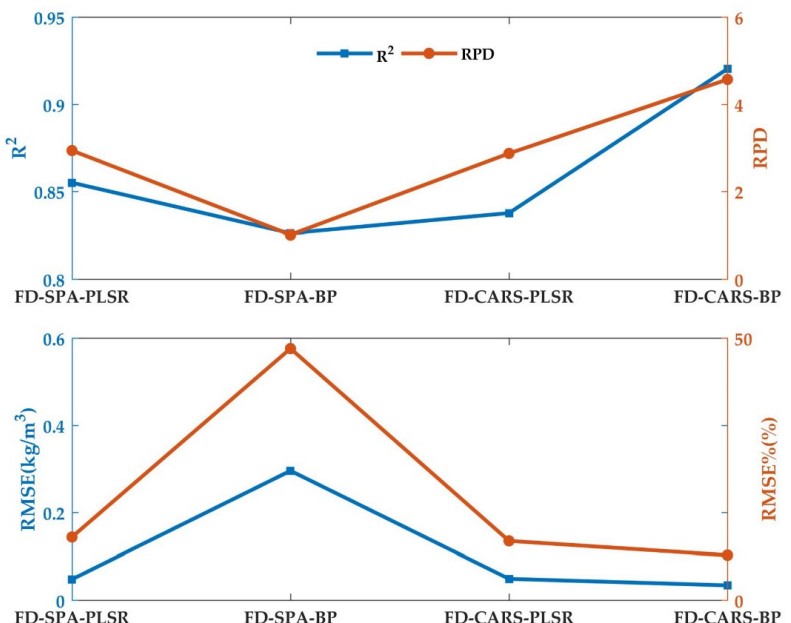

**Figure 12.** Comparison of inversion results of airborne hyperspectral data in 2016.

## 5. Discussion

### 5.1. Construction of Inversion Model

Six wavelengths of 655 nm, 660 nm, 840 nm, 860 nm, 870 nm, and 859 nm were selected to construct single-band empirical models with the measured SSSC, including linear, exponential, logarithmic, multiplicative power, and quadratic polynomial models. $R^2$, RPD, RMSE, and RMSE% were calculated to evaluate the accuracy of the inverse model and compare with the accuracy of the FD-CARS-BP model proposed in this paper, as shown in Table 6. Compared with the commonly used satellite visible and near-infrared wavelengths, where 655 nm and 870 nm are located in bands 4 and 5 of Landsat 8 OIL, respectively; 660 nm and 840 nm correspond to bands 3 and 4 of Landsat TM, respectively; 860 nm and 870 nm correspond to bands 2 and 16 of MODIS, respectively, which are the commonly used wavelengths for quantitative sand concentration inversion; 859 nm is the band with the largest correlation coefficient in hyperspectral images. It can be found that the accuracy of the FD-CARS-BP model has been significantly improved compared with the single-band model. Because the number of bands in multispectral remote sensing is small, the spectral resolution is low, and the spectral range of each band is long, which cannot express the changes in spectral information more finely, while the narrow bands

in hyperspectral images can effectively solve this problem, and the BP neural network model constructed based on the feature bands extracted by CARS can derive quantitative estimations of the SSSC more accurately.

**Table 6.** Comparison of different inversion models.

| Variable | Model | $R^2$ | RPD | RMSE (mg/L) | RMSE% |
|---|---|---|---|---|---|
| 655 nm | $y = 10093x - 115.67$ | 0.7930 | 1.9634 | 96.6928 | 47.71% |
| | $y = 8.8443e^{78.31x}$ | 0.9821 | 3.4376 | 81.8898 | 33.37% |
| | $y = 126.13\ln(x) + 677.1$ | 0.3965 | 0.9219 | 169.7026 | 98.12% |
| | $y = 280010x^2 - 6409.2x + 43.773$ | 0.9665 | 5.3117 | 39.9233 | 18.21% |
| 660 nm | $y = 10173x - 113.17$ | 0.7969 | 1.9876 | 95.7472 | 47.14% |
| | $y = 9.1108e^{78.596x}$ | 0.9828 | 3.3746 | 84.1994 | 34.16% |
| | $y = 124.38\ln(x) + 673.57$ | 0.3937 | 0.9883 | 170.6553 | 93.01% |
| | $y = 279307x^2 - 6112.1x + 41.65$ | 0.9682 | 5.4470 | 38.9028 | 17.74% |
| 840 nm | $y = 22336x - 28.066$ | 0.9703 | 5.3603 | 37.8763 | 17.35% |
| | $y = 24.442e^{140.37x}$ | 0.8748 | 1.7822 | 232.9018 | 84.36% |
| | $y = 160.08\ln(x) + 1004.9$ | 0.7059 | 1.5699 | 115.0457 | 56.19% |
| | $y = 202027x^2 + 16921x - 8.335$ | 0.9870 | 7.8004 | 26.3551 | 12.01% |
| 860 nm | $y = 25350x - 13.708$ | 0.9836 | 6.84 | 29.4738 | 13.55% |
| | $y = 27.664e^{155.33x}$ | 0.8621 | 1.7759 | 232.1537 | 86.01% |
| | $y = 153.96\ln(x) + 1011.2$ | 0.7574 | 1.6222 | 105.2444 | 49.97% |
| | $y = 105858x^2 + 22906x - 6.5177$ | 0.9885 | 8.0156 | 25.3042 | 11.61% |
| 870 nm | $y = 27158x - 7.3402$ | 0.9776 | 6.1185 | 33.267 | 15.25% |
| | $y = 32.182e^{158.01x}$ | 0.8496 | 1.735 | 242.3157 | 89.10% |
| | $y = 162.45\ln(x) + 1071.2$ | 0.6016 | 1.5411 | 161.5067 | 92.94% |
| | $y = 91181x^2 + 25205x - 2.011$ | 0.9823 | 6.8874 | 29.7421 | 13.64% |
| 859 nm | $y = 24537x - 12.381$ | 0.9792 | 6.1074 | 34.7046 | 16.16% |
| | $y = 26.63e^{160.09x}$ | 0.8569 | 0.8011 | 264.5516 | 92.54% |
| | $y = 146.04\ln(x) + 957.7$ | 0.7286 | 1.8937 | 111.9162 | 56.06% |
| | $y = 30674x^2 + 23845x - 10.34$ | 0.9812 | 6.3617 | 33.3162 | 15.50% |
| 689 nm/737 nm | $y = -312.32x + 909.04$ | 0.6771 | 1.7564 | 120.6730 | 55.11% |
| | $y = 15515e^{-2.181x}$ | 0.9811 | 6.3245 | 33.5115 | 15.42% |
| | $y = -740.5\ln(x) + 785.62$ | 0.7872 | 2.1519 | 98.4939 | 44.54% |
| | $y = 189.27x^2 - 1213.6x + 1923.7$ | 0.9091 | 3.1233 | 67.8582 | 30.19% |
| 717 nm − 400 nm | $y = 19738x - 31.298$ | 0.9453 | 4.2041 | 50.4119 | 23.78% |
| | $y = 24.99e^{120.42x}$ | 0.9650 | 2.3001 | 92.1450 | 39.64% |
| | $y = 195.92\ln(x) + 1144.6$ | 0.7858 | 2.1578 | 98.2221 | 47.83% |
| | $y = 291334x^2 + 10938x + 5.2895$ | 0.9727 | 5.7809 | 36.6645 | 17.19% |
| 859 nm + 859 nm | $y = 12795x - 17.05$ | 0.9792 | 6.1712 | 32.8574 | 14.95% |
| | $y = 26.932e^{78.775x}$ | 0.8610 | 1.7498 | 240.0633 | 87.32% |
| | $y = 156.48\ln(x) + 912.48$ | 0.7286 | 1.6072 | 110.4144 | 52.91% |
| | $y = 35185x^2 + 11178x - 7.4145$ | 0.9868 | 7.5416 | 27.1006 | 12.30% |
| $\frac{689\ nm - 737\ nm}{689\ nm + 737\ nm}$ | $y = -1758.5x + 847.52$ | 0.8232 | 2.0634 | 89.6017 | 41.44% |
| | $y = 6866e^{-11.42x}$ | 0.9740 | 5.4736 | 42.9276 | 19.95% |
| | $y = -601\ln(x) - 426.44$ | 0.9192 | 3.0110 | 62.5053 | 28.79% |
| | $y = 4534.4x^2 - 4988.2x + 1375.2$ | 0.9647 | 4.0573 | 46.0922 | 21.07% |
| | MLR | 0.9788 | 6.8815 | 30.8272 | 15.12% |
| | PLSR | 0.9697 | 5.6096 | 37.7826 | 17.71% |
| | FD−CARS−BP | 0.9947 | 12.5453 | 16.5801 | 8.08% |

Because the reflectance of a single band is too weak to reflect the spectral information of different SSSCs comprehensively, the dual-band combination model is also a common model form in the study of SSSC inversion. The combination of any two bands in the hyperspectral data is selected to construct four hyperspectral water body indices, namely, difference water index (DWI), ratio water index (RWI), normalized difference water index (NDWI), and addition water index (AWI), and the correlation coefficient matrix is calculated by correlating the water body indices with the measured SSSC. The correlation coefficient matrices of SSSC are symmetric matrices; thus, for the DWI, AWI, and NDWI correlation coefficient matrices, only the lower triangular matrix is selected to represent them, as

shown in Figure 13. It can be seen that the combinations of bands with high correlation coefficients all contain NIR bands; therefore, for regions with high SSSCs such as the Yangtze River estuary and adjacent seas, NIR bands are sensitive to changes in SSC, which is basically consistent with the conclusions of existing studies [58]. For RWI, the maximum value of correlation coefficient is 0.9403, corresponding to wavelengths of 689 nm and 737 nm; for DWI, the maximum value of correlation coefficient is 0.9765, corresponding to wavelengths of 400 nm and 717 nm; for AWI, the maximum value of correlation coefficient is 0.9842, corresponding to wavelengths of 859 nm and 859 nm; for NDWI, the linear model, exponential model, logarithmic model, and quadratic polynomial model were constructed with the highest correlation coefficient as the independent variable and SSSC. The $R^2$, RPD, RMSE, and RMSE% were calculated to evaluate the accuracy of each model, all of which were compared with the accuracy of FD-CARS-BP model, as shown in the table. It can be observed that the accuracy of the FD-CARS-BP model is still significantly higher than that of the combined two-band model.

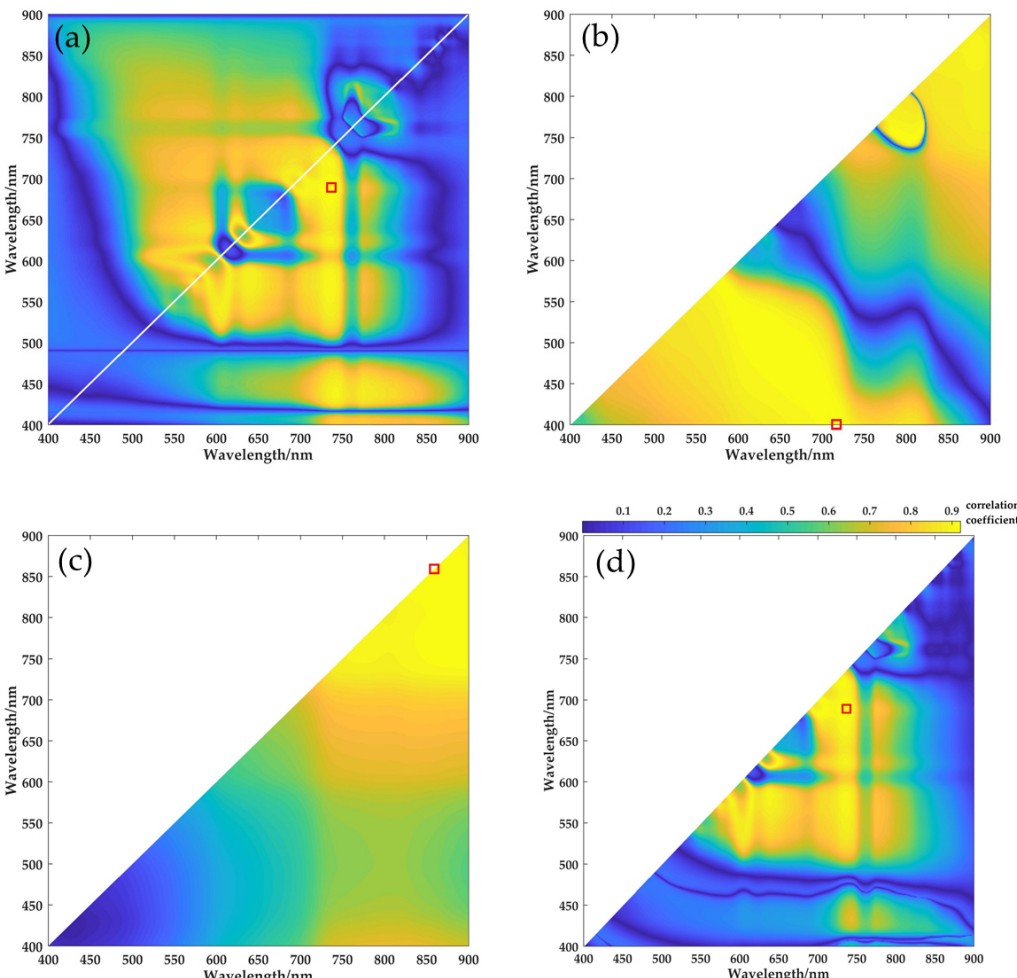

**Figure 13.** Correlation matrix. (**a**) Correlation coefficient of DWI, (**b**) correlation coefficient of RWI, (**c**) correlation coefficient of AWI, (**d**) correlation coefficient of NDWI. The small red squares in this figure refers to the band with the largest correlation coefficient, that is, the characteristic band.

Whether it is a single-band model or a two-band combined model, it is still a one-dimensional model by nature, and the one-dimensional function often does not explain the dependent variable to the same extent as the multivariate function. Furthermore, the multivariate model is also one of the models commonly used in the inversion studies of suspended sand concentration. Therefore, a multiple linear regression model (MLR) is constructed based on the feature bands extracted by CARS, and four indicators, $R^2$, RPD,

RMSE, and RMSE%, are calculated and compared with the PLSR model and BP neural network model for accuracy, and the results are shown in Table 6. It can be seen that the accuracy of the FD-CARS-BP model proposed in this study is still higher than that of other models, and its $R^2$ value is as high as 0.99, its RPD is greater than 12, its RMSE is less than 16 mg/L, and its RMSE% is only 8.08%, all of which have more obvious advantages.

*5.2. The Validation of the Model Framework*

The model framework of this study is proposed based on quantitative experiments simulating water bodies with different suspended sand concentrations. The measured spectral curves of water bodies with different SSSCs can show the trend of remote sensing reflectance with the change in SSSC, and then guide the selection and determination of hyperspectral characteristic wavebands. Compared with the traditional single-band model, dual-band combined model, and multiple linear regression model, this model can effectively improve the accuracy of the inversion of SSSC and obtain the distribution characteristics of SSSC more accurately. Applying the framework to the airborne hyperspectral suspended sand concentration monitoring in the typical area of the North Port of the Yangtze River estuary, it is observed that the accuracy of the model is still significantly higher than that of other model combinations, which further verifies the validity of the model, and can realize the high-precision inversion study of suspended sand concentration in the typical local area of the Yangtze River estuary.

*5.3. The Limitations of the Model Framework*

The model framework also has some limitations. First, regarding the quantitative experiments of different SSSCs in simulated water bodies, the water bodies used during the experiments in this paper were tap water, and only the effects of different SSSCs on the spectral reflectance of water bodies were considered. The mechanisms of the effects of salinity, chlorophyll, and other organic matter on the spectral determination of water bodies will be discussed in a subsequent study. Secondly, in the dataset used for the quantitative experiments in this paper, the range of suspended sand concentration is 3.62–682.06 mg/L. For the water bodies with suspended sand concentrations higher than 682.02 mg/L, the spectral characteristics need to be further investigated. Moreover, regarding the airborne hyperspectral SSSC monitoring, the experimental data collection time is concentrated in the dry water period, and whether the model proposed in this paper can be applied to the abundant water period needs to be further verified. Under the influence of runoff, sand transport, tides, waves, wind systems, climate, geological structures, and human activities, the distribution and dispersion patterns of SSSC are exceptionally complex. The water spectrum varies greatly in different seas, and a unified SSSC inversion regionalization model is yet to be constructed. With the continuous development of hyperspectral technology, water quality parameter inversion based on hyperspectral images can be further explored in the future, and the de-characterization method of sensitive bands can be continuously explored to optimize the inversion model and further improve the accuracy of water quality parameter inversion. Meanwhile, for SSSC, the influence of tides, flow velocity, flow direction, and other factors on its spatiotemporal distribution can be explored to obtain finer regularity characteristics.

## 6. Conclusions

In this study, we propose a framework for the inversion of SSSC based on hyperspectral remote sensing. Our study demonstrates the potential of static load hyperspectral remote sensing-based techniques for monitoring SSSC in the Yangtze estuary region. We investigated a process-based approach, the FD-CARS-BP framework, to analyze wavelengths sensitive to SSSC, and constructed a reliable SSSC inversion model. In the quantitative experiments of the flume, the spectral changes in the water column at different SSSCs were analyzed and compared to those of multiple spectral transformation methods. Second, two different waveband selection methods were implemented, and then, based on comparing

SPA, CARS, and correlation analysis methods, the FD-CARS algorithm was proposed to extract feature bands to invert SSSC. The use of the BP inversion model after the FD-CARS algorithm significantly reduced the number of bands and improved the inversion accuracy of the SSSC in quantitative experiments. Finally, the FD-CARS-BP model was applied to an actual SSSC inversion experiment of airborne hyperspectral remote sensing in the Yangtze estuary in 2016, which further verified the reliability of the framework proposed in this study. This framework provides a simple and practical method for inverting SSSC in the Yangtze estuary. The purpose of the quantitative simulation experiment was to consider only the effect of SSSC changes on light reflectance; therefore, the water body used in the experiment was tap water, and the mechanisms of the effects of other organic substances, such as chlorophyll and CDOM, on the spectral characteristics of the water body will be considered in a subsequent study. In the future, we hope that the framework can be combined with classification filters to automate the identification process.

**Author Contributions:** Methodology, K.L. and H.L.; validation, J.W., C.G., C.Q. and Y.P.; formal analysis, W.Z. and Z.Q.; investigation, H.X.; writing—original draft preparation, K.L.; writing—review and editing, H.L.; funding acquisition, K.L. All authors have read and agreed to the published version of the manuscript.

**Funding:** This research was funded by Shanghai Ocean Bureau Research Project (Shanghai 2019-05).

**Data Availability Statement:** The data presented in this study are available on request from the corresponding author. The data are not publicly available due to the data relate to real geographic coordinates, and other possibilities for inversion of the water environment in the Yangtze estuary.

**Acknowledgments:** Thanks to the Shanghai Institute of Technology Physics, Chinese Academy of Sciences for the air flight support.

**Conflicts of Interest:** The authors declare no conflict of interest.

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
