# Peer review of "Quantitative Inversion Method of Surface Suspended Sand Concentration in Yangtze Estuary Based on Selected Hyperspectral Remote Sensing Bands"

_sustainability, doi:10.3390/su142013076_

Round 1

Reviewer 1 Report

Point 1. Clarify the reasons for selecting the size of the plastic tank for the experiment of SSSC determination (line 194);

Point 2. As stated in the abstract "the distribution of SSSC in the estuary area is extremely complex because of its complex and variable topography and hydrodynamic condition". In this research, the airborne hyperspectral experiment was carried out at one specific area upstream of the northern port of the Yangtze Estuary (Figure 2) and only once time/campaign (during the dry water periods on March 26, 2016). So can the outcome of this research be applied or represented for the whole study area (Yangtze estuary) and for all period (with different hydrodynamic condition, especially in the flood season with large amount of sedimentation comes from the upstream of the river) 

Other comment: Please check grammar - there are still a few minor errors, e.g. Line 224

Reviewer 2 Report

In this paper, a remote sensing estimation algorithm of suspended matter concentration in the Yangtze Estuary is constructed by using hyperspectral remote sensing data. Overall, the research theme of this paper well-fits the scopes of Remote Sensing. Reviewer appreciate all of author's effort for the manuscript. The paper is innovative to a certain extent, but there are still many problems that need further revision.

Major comments

(1)   The paper uses the neural network model to construct a model for estimating the concentration of TSM in the characteristic band, but the paper has a total of 41 samples, is it not enough for the training samples? And please introduce the training sample size and the validation sample size.

(2)   In introduction, the structure logic is confused, and the introduction of research progress is not a simple introduction to the existing literature, which needs to be further sorted out.

(3)   At present, there are many researches on remote sensing estimation of suspended sediment. In the Yangtze Estuary where suspended sediment is high, high-precision estimation can be achieved by a single band, and the purpose of remote sensing is to be applied to practice. Therefore, the research significance of this paper needs to be further elaborated.

(4)   The model proposal of the paper should be compared with other existing algorithms to further reflect the advantages of the model.

(5)   The discussion section of the paper should include a discussion of the applicability and limitations of the model

Minor Comments

(1)   Figures 2 and 3 can be merged and a map of China added.

(2)   Line 31, RMSE should have units, the full text needs to be modified.

(3)   Units should be added to Rrs in Figure 5.

(4)   The line chart and bar chart in the paper are suggested to be drawn with professional software.

       The full text of the formula size and style need to be unified.

Reviewer 3 Report

You should revise the title to make it shorter in length and more precise to reflect your basic contribution. Because you are not proposing any novel method for band selection but rather using existing methods in solving your problem. For example: “Quantitative inversion method of surface suspended sand concentration in Yangtze estuary based on selected hyperspectral remote sensing bands”---makes it more and accurately informative.

 Please revise line #23 -#27, line#178-#183 to segment the long sentence into several sentences.

Revise the sentence line#186-#189 : “Finally, the constructed framework was applied to the 2016 airborne hyperspectral simultaneous monitoring experiment in the Yangtze River estuary to further validate its effectiveness in the inversion process of the SSSC in the Yangtze River estuary and to provide methodological support for the hyperspectral remote sensing inversion of  the SSSC.

 In Table-1:

1.      Check the first three spectral parameters as they do not seem to comply technically

2.      Check the value of “Integration time”. Should it be in nm unit?

 Revise: “When the values were stable, the spectral characteristics of the SSSC were measured using the above  water surface measurement method...what do mean by above?

 Check proper format for caption in Figure 1

In Table-4: check the first three spectral parameters as they do not seem to comply technically

 Revise:” The raw spectra were preprocessed to include five spectral transformation forms, as  shown in Table 4.

 Table 4: Check the equation for “Square Root”—there is no square root operation

 In section 3.2 (overall try to reduce unnecessary texts): I think the SPA or CARS algorithms need not be elaborately described. Subsection 3.3 is required to be shortened as it is not necessary to describe the basic operations of BP neural net.  

Revise: line#389-line#391

 Section 5 (Discussion): The contents are not appropriate to include in the discussion part.  They should be distributed in the Materials and Methods, and Result section.

 Use any of the metrics of RMSE and RMSE% as they represent the same information.

A result comparison with other state-of-the-art methods should be included.

The quality and readability of images should be increased.  

The scientific style of writing is not of a good standard. In general, you should rearrange the current contents of sections 2,3,4, and 5 into Materials, Methods, and Result sections.

A section of Discussion can be added to discuss the miscellaneous subjects about the experiment, and results. Also, the limitations and critical parameters of the study can be discussed in the Discussion Section.   

Round 2

Reviewer 1 Report

Accept

Reviewer 2 Report

The article has been carefully revised to solve my doubts, and I propose to accept it